# Investigation of Conditions for Capture of Live *Legionella pneumophila* with Polyclonal and Recombinant Antibodies

**DOI:** 10.3390/bios12060380

**Published:** 2022-05-31

**Authors:** Lucas Paladines, Walid M. Hassen, Juliana Chawich, Stefan Dübel, Simon Lévesque, Jan J. Dubowski, Eric H. Frost

**Affiliations:** 1Laboratory for Nano-Technologies and Nano-Characterization (LN2)-CNRS UMI-3463, Interdisciplinary Institute for Technological Innovation (3IT), Université de Sherbrooke, 3000 Boul. de l’Université, Sherbrooke, QC J1K 0A5, Canada; lucas.antonio.paladines.alvarez@usherbrooke.ca (L.P.); mohamed.walid.hassen@usherbrooke.ca (W.M.H.); juliana.chawich@usherbrooke.ca (J.C.); jan.j.dubowski@usherbrooke.ca (J.J.D.); 2Department of Microbiology and Infectiology, Faculty of Medicine and Health Science, Université de Sherbrooke, 3001 12th Avenue North, Sherbrooke, QC J1H 5N4, Canada; simon.levesque@usherbrooke.ca; 3Department of Electrical and Computer Engineering, Faculty of Engineering, Université de Sherbrooke, 2500 Boul. de l’Université, Sherbrooke, QC J1K 2R1, Canada; 4Department of Biotechnology, Technische Universität Braunschweig, Spielmannstrasse 7, 38106 Braunschweig, Germany; s.duebel@itubs.de; 5CIUSSS de l’Estrie-Centre Hospitalier Universitaire de Sherbrooke, 3001, 12th Avenue North, Sherbrooke, QC J1H 5N4, Canada

**Keywords:** *Legionella pneumophila*, bacteria detection, recombinant antibody, polyclonal antibody, biosensor, qPCR, 96-well microtiter plate

## Abstract

Since *Legionella pneumophila* has caused punctual epidemics through various water systems, the need for a biosensor for fast and accurate detection of pathogenic bacteria in industrial and environmental water has increased. In this report, we evaluated conditions for the capture of live *L. pneumophila* on a surface by polyclonal antibodies (pAb) and recombinant antibodies (recAb) targeting the bacterial lipopolysaccharide. Using immunoassay and PCR quantification, we demonstrated that, when exposed to live *L. pneumophila* in PBS or in a mixture containing other non-target bacteria, recAb captured one third fewer *L. pneumophila* than pAb, but with a 40% lower standard deviation, even when using the same batch of pAb. The presence of other bacteria did not interfere with capture nor increase background by either Ab. Increased reproducibility, as manifested by low standard deviation, is a characteristic that is coveted for biosensing. Hence, the recAb provided a better choice for immune adhesion in biosensors even though it was slightly less sensitive than pAb. Polyclonal or recombinant antibodies can specifically capture large targets such as whole bacteria, and this opens the door to multiple biosensor approaches where any of the components of the bacteria can then be measured for detection or characterisation.

## 1. Introduction

*Legionella pneumophila* is a gram-negative bacterium that is ubiquitous in water sources, environmental or man-made [1]. Sixteen serogroups of *L. pneumophila* are defined by the various components of the surface, but principally due to the variability of the lipopolysaccharide (LPS) [2]. In the environment, *L. pneumophila* is able to survive, despite its nutritional dependence on cysteine and iron, by having a high resilience in water [2,3] and by infecting protozoa, especially amoeba [4]. When engulfed by amoeba or human macrophages, they prevent lysosome fusion after phagocytosis and use nutriments of the host to divide and eventually cause host cell death [5]. This ability to survive intracellularly makes them pathogenic for humans, with different levels of pathogenicity depending on the individual’s immunologic strength and the pathogenic potential of the microorganism. The severity of the infection varies from asymptomatic, Flu-like symptoms, Pontiac fever, to Legionellosis and even death [6]. Even though 16 serogroups exist for *L. pneumophila*, 75% of infections, including outbreaks, are caused by *L. pneumophila* ssp 1 [7,8]. Due to its ubiquitous presence and proliferation in man-made water systems, Legionellosis outbreaks usually originate from water sources, especially those in industry. Since the first reported *L. pneumophila* epidemic that occurred in 1976 [9], outbreaks have occurred in man-made water [10] structures, with the following two vibrant examples occurring recently: the outbreaks in New York City in 2015 (138 cases and 16 deaths) [11] and in Quebec City in 2012 (182 infected and 13 deaths) [12]. In both occurrences, cooling tower water (CTW) was the source as follows: *L. pneumophila* infected microdroplets were dispersed from large structures such as skyscrapers or industrial buildings. A CTW environment provides suitable conditions for the growth of *L. pneumophila* in bacterial biofilms and protozoa. Driven by the 2012 Quebec City epidemic, new intervention guidelines for *Legionella* were implemented in 2015 in Quebec province [13], requiring that CTW must be monitored for the presence of *Legionella* at least once a month in Quebec. In recent legislation, similar guidelines have been established for New York City and State [14]. Even if PCR can provide results in 2–3 h after sample collection and transport to a laboratory, culture on agar plates remains necessary to validate the PCR results, and this usually requires a 5-day incubation but up to 14 days of obligatory plate observation before the culture can be considered negative [13]. Indeed, overestimation of positivity by PCR is possible due to the detection of DNA from non-viable and even dead bacteria, and confirmation by culture remains necessary [13]. Both culture and PCR require highly qualified personnel and time, which, during an epidemic, can cost lives. Therefore, the development of biosensors could provide fast, quasi-real-time, and even prospective control of *L. pneumophila* and thus avoid or curtail future outbreaks [15].

Several biosensing platforms have been investigated for the detection of *L. pneumophila*, such as quartz crystal microbalance (QCM) [16], surface plasmon resonance (SPR) [16,17] and electrical impedance spectroscopy (EIS) [16]. Based on the photonic characteristics of digital photocorrosion (DIP) of GaAs/AlGaAs heterostructures functionalized with antibodies, our group [18] has developed DIP based biosensors allowing the detection of *Escherichia coli* K12 at 10^3^ CFU/mL [19,20] and heat-inactivated *L. pneumophila* ssp 1 at 10^3^ CFU/mL [21,22].

The sensitivity of such biosensors is highly dependent on their ability to specifically capture the targeted bacteria, which is very dependent on the molecules used for capture. Antibodies have been shown to enhance the specificity of the reaction, as polyclonal antibodies provide stronger antibody-target bonds or affinity, which might improve sensitivity but could give rise to more false positives [23,24,25]. However, monoclonal and especially recombinant antibodies, have also shown increased reproducibility [16]. Most previous studies have used antibodies to capture individual proteins and lipopolysaccharides or alternatively, to agglutinate larger entities such as whole bacteria or viruses. Capture and retention of large entities on a surface has not been extensively examined. This may prove useful for SPR, DIP, and other detection strategies, which could benefit from the presence of large targets. Furthermore, living bacteria could be further characterised to provide a second level of authentication or additional information such as antimicrobial sensitivity. As an alternative to antibodies, we have also employed capture with short-ligand and sandwich antimicrobial peptide architectures that allowed detection of *L. pneumophila* at 1–2 × 10^2^ CFU/mL [26].

In the present study, we report the performance of a recombinant antibody (recAb) specific to *L. pneumophila* ssp 1 LPS [21] as compared to polyclonal Ab (pAb) to capture whole, live bacteria on a surface. The sensitivity, specificity, and reproducibility of pAb and recAb to capture *L. pneumophila* were evaluated to select an optimal strategy for biosensing.

## 2. Materials and Methods

### 2.1. Materials

*L. pneumophila* ssp 1 expressing Green Fluorescent Protein (GFP *L. pneumophila*), a strain transformed with an Isopropyl B-D-1-thiogalactopyranoside (IPTG)-inductive plasmid producing GFP maintained by chloramphenicol was kindly provided by Prof. Sébastien Faucher (McGill University, Montréal, QC, Canada). *L. pneumophila* ssp 5 [27], *E. coli*, *Pseudomonas aeruginosa*, *Bacillus subtilis*, and *Staphylococcus aureus* were provided by the department of microbiology of the Centre Hospitalier Universitaire de Sherbrooke (Sherbrooke, QC, Canada). Buffered charcoal yeast extract (BCYE) agar was obtained from Becton, Dickinson, and Company (Bergen County, MD, USA). Luria-Bertani (LB) plates and LB broth were obtained from EMD chemicals (Darmstadt, Germany). L-cysteine, Iron (III)-pyrophosphate, ethylenediaminetetraacetic acid (EDTA), ethanolamine, *Legionella* enrichment broth and Sodium dodecyl sulfate (SDS) were purchased from Sigma-Aldrich (Oakville, ON, Canada). Rabbit polyclonal anti-*Legionella pneumophila* Ab (pAb) was purchased from Virostat, Inc. (Portland, ME, USA). The human recombinant scFv antibody fragment against *Legionella* LPS was obtained by antibody phage display and fused to human Fc to form PHK121-H2h [28]. Costar 3590 96-well microtitre plates were obtained from Corning Inc. (Corning, NY, USA). Phosphate-buffered saline (PBS) and Tris-HCl were purchased from Amresco (Solon, OH, USA) and Invitrogen (Carlsbad, CA, USA), respectively. Triton X-100 and Tween-20 were obtained from Fisher scientific (Branchburg, NJ, USA). All solutions were made in PBS, except Ethanolamine which was dissolved in sterile deionized (DI) water. Molecular grade water was obtained from Wisent Inc. (Saint-Jean-Baptiste, QC, Canada). The Plate shaker (PMX-01E) and the Ultrasonicator Branson 200 (40 kHz) were purchased from Fujirebio inc (Shanghai, China) and Branson Ultrasonics corp. (Brookfield, CT, USA), respectively. For qPCR, the complete master mix was obtained from the protein purification service of the Department of microbiology and infectiology (Université de Sherbrooke, QC, Canada) and PCR system LightCycler^®^ 480 II (Roche diagnostics, Indianapolis, Indiana, USA) was used for all samples. Primers and probes were obtained from Integrated DNA technologies (Coralville, IA, USA).

### 2.2. Bacterial Culture

In preliminary experiments, we occasionally observed overgrowth with strains resembling *L. pneumophila*, so we employed a strain-producing GFP to provide a double guarantee of the identity of the bacteria tested as follows: it does not grow on BCYE plates without L-Cys, and colonies appear green on BCYE plates overlaid with chloramphenicol, which induces GFP production. *L. pneumophila* ssp 5 was also confirmed by its lack of growth on BCYE plates without L-Cys. *L. pneumophila* was plated and grown on BCYE agar plates coated with chloramphenicol for 4–5 days at 37 °C. An isolated colony was inoculated in *Legionella enrichment broth* (L.ER) supplemented with iron-pyrophosphate and L-cysteine and incubated for 24–48 h at 37 °C with shaking (~100 rpm). *E. coli, B. subtilis* and *S. aureus* were cultivated on LB plates overnight at 37 °C (O/N, 37 °C) then an isolated colony was inoculated into LB broth and grown (O/N, 37 °C) with shaking (~100 rpm). Exponential phase cultures of *L. pneumophila* were used as their outer membranes varied only slightly from viable but non-culturable *L. pneumophila* found in nutriment lacking environments [2], such as CTW. They were confirmed by having neutral pH [29] and no brown colour [30] (see Section A.1). The same protocol was applied to *P. aeruginosa*, but at a temperature of 25 °C. The bacterial concentrations were quantified by optical density measurements at 600 nm (OD_600 nm_). OD_600 nm_ of 0.1 was 6 × 10^7^ CFU/mL for *L. pneumophila*, 2.5 × 10^8^ CFU/mL for both *E. coli* and *S. aureus* [31], and 2.5 × 10^7^ CFU/mL for both *B. subtilis* [32] and *P. aeruginosa* [33], respectively. All bacterial suspensions were prepared in PBS.

### 2.3. Immuno-Capture of Live Bacteria on 96 Well Plates

As a proxy for capture of bacteria with Ab on GaAs plates or other biosensor surfaces, *L. pneumophila* was captured with Ab on 96 well Costar plates. Ab solutions at 100 µg/mL in PBS were incubated in the plate wells for 1 h at room temperature (RT). Following adsorption, the following two blocking steps were carried out: first with BSA 2% in PBS and then with ethanolamine (pH 8, 1 M) in DI water. Both blocking solutions (200 µL/well) were incubated in the wells for 1 h at RT. After blocking, the wells were washed 3 times with PBS. For bacterial capture, 100 µL of *L. pneumophila* suspensions were added to each well (1 h, RT). Finally, 3 successive washes were made with PBS + Tween 0.05% (PBST). The effect of 0.05% Tween in PBS on the viability of *L. pneumophila* was evaluated (see Section A.2). To test the impact of Tween on reducing non-specific and antibody-specific capture, experiments were performed with or without Tween 0.05% in the bacterial washing step. All described incubations were performed with shaking (~500 rpm). For background level wells, the step of adding Ab solution was replaced by adding PBS instead, while all other steps remain unchanged.

### 2.4. Effect of Antibody Pre-Sonication on Bacterial Capture

In order to test the effect of sonication on possible rAb aggregation, 100 µL aliquots of Ab were pre-sonicated in a sonication bath at 4 °C in glass tubes for various times to reduce possible Ab aggregation before adding the Ab solution to the wells. This step was performed in a walk-in cold room at 4 °C. The ensuing blocking and washing steps remained the same as described in the previous section.

### 2.5. Direct DNA Extraction

An adaptation of the EtNa DNA extraction [34] protocol was chosen for direct DNA extraction of different microorganisms from Costar plates because of the limited material requirements and minimal loss of DNA. The DNA suspension solution from the *EtNa* protocol (Triton X-100 1%, Tween 20 0.05%, Tris-HCL 50 mM, EDTA 0.1 mM, pH 8) was used to detach bacteria captured on the plate wells. After adding 200 µL of the DNA extraction solution to the wells, the plate was heated to 85 °C for 30 min in a water bath, then cooled on ice for 5 min, quickly centrifuged and placed at −15 °C O/N. After thawing, the samples were aliquoted into microtubes and used for PCR.

### 2.6. Polymerase Chain Reaction (qPCR)

A 2 µL sample of the direct DNA extract was added to 8 µL of amplification solution in a LightCycler PCR plate as follows: 0.25 µM MIP forward and reverse primers, 0.15 µM MIP probe, (and/or 0.1 µM 16S rDNA forward and reverse primers, 0.075 µM 16S rDNA probe), 5 µL of 2X Taqman master mix (produced by the protein and purification service of the University of Sherbrooke, Sherbrooke, QC, Canada), completed to 8 µL with Mol Grade H_2_O. MIP gene primers and probes were employed for specific detection of *L. pneumophila* [35], as shown in Table 1. For universal amplification, 16S ribosomal DNA (16S rDNA) primers and probes, adapted from the *CDC* protocol [36], were added, as shown in Table 1. PCR program (activation: 95 °C-15 s; 50 amplification cycle: 95 °C-15 s, 60 °C-1 min) was optimized for the primers and probes. After amplification, the LightCycler programme determined the cycle threshold (Ct) of detection.

### 2.7. Bacteria Quantification Curve

Triplicate 100 µL volumes containing concentrations from 10^3^ to 10^8^ CFU/mL of bacteria were added to a Costar plate in 1 log_10_ increments to obtain 6 concentrations of *L. pneumophila*. The addition of 100 µL of 2X concentrated DNA extraction solution to the wells resulted in concentrations from 5 × 10^2^ to 5 × 10^7^ CFU/mL of *L. pneumophila*. DNA extraction of the bacteria added to the Costar plate was carried out as described above and amplified by qPCR. Quantification curve wells were included in each experiment to assess the variability of extraction and amplification of positive controls and to obtain sufficient replicates for the quantification curve (Figure 1). Ct values were plotted against bacterial concentrations to give a linear calibration curve. Experimental Ct values obtained from experiments were employed to quantify the number of immobilised bacteria by interpolation with the calibration curve.

## 3. Results and Discussion

### 3.1. Calibration Curve

Following DNA extraction of *L. pneumophila* from suspensions with concentrations ranging from 5 × 10^2^ to 5 × 10^7^ CFU/mL, qPCR amplification was carried out, and Ct values were determined with two sets of primers and probes [37]. Standard curves generated by linear regression revealed an inverse linear correlation between Ct and the concentration of *L. pneumophila* (Figure 1).

Lower concentrations were not tested as the Ct for 5 × 10^2^ CFU or less showed a plateau, indicating that the detection limit of the system had been reached. This is not surprising as, theoretically, there would be only one bacterium in a 2 µL aliquot from a bacterial solution with 5 × 10^2^ CFU/mL, so lower concentrations of bacteria might be expected to generate negative qPCR results. The difference in Ct values between the two curves for the same bacterial concentration was due to the presence of three copies of the 16S rDNA sequence [38] but only a single copy of the MIP gene per *Legionella* bacteria [39,40]. The calibration curve was used to estimate the number of immobilised bacteria in the plate wells, with the MIP gene curve used for *L. pneumophila* and the 16S gene curve for *L. pneumophila* and other bacteria. Each capture test included calibration curve wells that also served as positive controls for the experiment.

### 3.2. Optimal Bacterial Concentration for Incubation in Ab-Coated Wells

We had previously observed that the optimal concentration of the solution of Ab to be deposited on GaAs or microplate surfaces was 100 µg/mL [41]. Hence, we used this concentration to evaluate the saturation level of the two Ab used to capture *L. pneumophila*.

To estimate the saturation level of an Ab-coated surface, Ab-coated wells were exposed to various concentrations of bacteria. It was presumed that in order to compare the ability of pAb and recAb to capture bacteria, it would be necessary to use bacterial concentrations that did not saturate the surface, as results would be expected to plateau with saturating concentrations and variation in conditions might not affect the values. The highest concentration tested was 10^9^ CFU/mL, and it provided the highest bacterial capture, as shown in Figure 2.

When suspensions at 10^9^ CFU/mL were added to wells coated with Ab, 2.2 × 10^6^ CFU/mL (0.22% of the input bacteria) were captured with pAb and 1.5 × 10^6^ CFU/mL (0.15%) with recAb, whereas without Ab, a non-specific capture of 3.6 × 10^4^ CFU/mL (0.0035%) was observed. Exposure to suspensions at lower concentrations resulted in specific capture of 4.1 × 10^5^ ± 2.2 × 10^5^ (0.41%) CFU/mL by pAb and 2.8 × 10^5^ ± 1.1 × 10^5^ (0.28%) CFU/mL by recAb at 10^8^ CFU/mL. The attachment of 3.0 × 10^4^ ± 1.7 × 10^3^ (0.30%) CFU/mL by pAb and 1.5 × 10^4^ ± 9.0 × 10^3^ (0.15%) CFU/mL by recAb were obtained by exposure to 10^7^ CFU/mL of *L. pneumophila*. Non-specific attachment of 5.8 × 10^3^ ± 4.8 × 10^3^ (0.0057%) and 1.0 × 10^3^ ± 1.6 × 10^2^ (0.01%) CFU/mL were obtained with exposure of 10^8^ CFU/mL and 10^7^ CFU/mL, respectively. The exposure to suspensions at lower concentrations (10^8^ CFU/mL and 10^7^ CFU/mL) resulted in lower bacterial retention than the 10^9^ CFU/mL, but similar capture percentages, indicating that capture was not less efficient. Hence, the 10^8^ and 10^7^ CFU/mL concentrations did not saturate the surface, and we do not have evidence that even 10^9^ CFU/mL saturated the surface. The percentage of capture on wells without Ab increased slightly with a decrease in the exposure concentration. This could be attributed to the saturation of non-specific capture sites on the wells with increasing concentrations of bacteria.

These results confirmed the use of 10^8^ or 10^7^ CFU/mL as exposure concentrations for experimentation as they did not saturate the surface. This would allow us to evaluate the variation in capture by Ab under different conditions. They also maintained values for non-specific detection of >10^3^ CFU/mL, which is within the range of the calibration curve.

### 3.3. Effect of Pre-Sonication of Antibodies on Bacterial Capture

Since only 0.1–1% of the total bacteria in the suspension were captured by Ab attached to the wells, we hypothesized that we lost some of the Ab capacity due to the presence of Ab aggregates. To reduce potential Ab aggregation, the Ab solutions were sonicated before adsorption to the wells to enhance bacterial capture. Figure 3 shows that different durations of sonication for both recAb and pAb did not significantly improve bacterial capture.

This result was expected for pAb due to its good solubility as it is produced by an organism for activity in solution [42]. Since the added scFv fragment may interfere with the solubility of recAb and cause self-aggregation [42], we checked this solution for aggregation. However, we did not observe that sonication improved capture efficiency, so we concluded that it was ineffective for breaking up aggregates or that aggregation did not pose a problem under our conditions.

### 3.4. Specificity of Polyclonal and Recombinant Antibodies

To evaluate Ab specificity, wells coated with pAb or recAb were exposed to different non-target bacteria (*E. coli*, *S. aureus*, *B. subtilis*, and *P. aeruginosa*). *L. pneumophila* ssp 5 was also used to test the selectivity towards other *Legionella* serogroups. Results based on the amplification of a universal 16S RNA showed that the average value of the capture of non-target bacteria to wells with pAb or recAb did not vary significantly from the background adhesion to the No-Ab wells, as shown in Figure 4, but the higher non-specific attachment of *B. subtilis* and *P. aeruginosa* to wells without Ab.

This indicated that Ab wells did not capture non-target bacteria significantly. However, *L. pneumophila* ssp 5 did show significant capture with recAb (2.0 × 10^3^ CFU/mL), albeit nearly 10 times less than the capture of *L. pneumophila* ssp 1 (1.5 × 10^4^ CFU/mL).

It was surprising that pAb did not capture more *L. pneumophila* ssp 5. Indeed, pAb is directed towards multiple components of *L. pneumophila* ssp 1, as indicated by the manufacturer, Virostat, who reported some cross-reactivity of this Ab with other serogroups of *L. pneumophila* [43,44]. However, this reactivity may not be towards components that allow the capture of *L. pneumophila*. LPS, or serogroup-specific LPS epitopes, may be the preferred target that allows the capture of these bacteria in the current experiments. When it was formulated, recAb was chosen to detect the LPS of ssp 1, and indeed, it was found highly specific for human isolates of serogroup 1 of *L. pneumophila* and did not detect an environmental serogroup 1, nor a serogroup 6 strain [22].

For further evaluation of specificity, mixtures of *L. pneumophila* at a concentration of 10^7^ CFU/mL were added together with competitor bacteria to evaluate the interference of other bacteria present in the solution with the adhesion of *L. pneumophila* to the surface of the wells as follows: *E. coli*; *S. aureus*; *L. pneumophila* sg5 at a concentration of 10^7^ CFU/mL (ratio 1:1) and *B. subtilis*; *P. aeruginosa* at a concentration of 10^6^ CFU/mL (ratio 10:1). Both sets of primers were used on all sample wells to evaluate the total number of bacteria captured on the surface vs. the number of *Legionella*. The results indicated that the presence of other bacteria did not interfere significantly with the attachment of *L. pneumophila* to the surface (Figure 5A,B, black bars).

It was observed that other bacteria did not adhere to the surface in the presence of attached *L. pneumophila*, as there was no increase in the 16S rDNA (grey columns) compared to the MIP (black columns) when other bacteria were added. The highest rate of non-specific attachment in wells not coated with Ab was observed with *P. aeruginosa* as follows: 1.2 × 10^4^ CFU/mL (Figure 5C), but this high non-specific attachment did not affect capture levels of *L. pneumophila* with Ab even when 16S primers and probe were used for evaluation (Figure 5A,B). This observation was also noted with all other bacteria mixtures. This result suggested that the Ab-*L. pneumophila* complex absorbed to the surface had acted as a blocking agent, preventing further binding of the non-target bacteria.

Considering specificity, both recAb and pAb showed excellent results when exposed to a variety of bacteria, making them both adequate choices for biosensor use.

### 3.5. Evaluation of Sensitivity and Reproducibility of Polyclonal and Recombinant Antibodies

From the various experiments (Figure 2, Figure 3, Figure 4 and Figure 5), both pAb and recAb showed comparable sensitivity under the same stringent conditions. pAb showed slightly higher numbers of captured bacteria compared to recAb. A compilation of capture levels and the standard deviation obtained for *L. pneumophila* captured with each Ab and quantified with 16S rDNA primers (from Figure 5) is presented in Figure 6.

These data indicate that pAb captured 4.32 × 10^5^ ± 2.72 × 10^5^ CFU/mL of live *L. pneumophila*, whereas recAb captured 2.99 × 10^5^ ± 1.64 × 10^5^ CFU/mL.

A slightly lower (70%) capture efficiency was observed for recAb compared to pAb; however, a slightly lower (60%) standard deviation was also observed for recAb compared to pAb, indicating that recAb captured *L. pneumophila* more consistently and more reproducibly than pAb. Several factors could contribute to these differences. RecAb is directed towards a single epitope of the *L. pneumophila* ssp 1 LPS, whereas pAb is probably composed of antibodies directed towards multiple epitopes of the LPS and perhaps other proteins as well. These other epitopes are not necessary for efficient capture but might contribute slightly to the strength of attachment to the target. On the other hand, reaction with other epitopes did not result in the capture of serogroup 5 *L. pneumophila* by pAb, perhaps because they are not as solidly anchored in the cell wall or membrane, or not as exposed as the LPS epitope targeted by recAb. It is interesting to note that recAb provided efficient capture, demonstrating that single epitopes of LPS can be a useful target for capturing bacteria and can be employed in biosensors.

It is to be expected that there will be variation between batches of pAb since they are produced in rabbits or other mammals where individual variation is observed. Since the production of a recAb does not require synthesis by a living vertebrate organism, it should be highly reproducible and provide identical specificity. The increased variation of results that we observed with pAb could not be attributed to batch-to-batch dissimilarity as we employed only a single batch for all of our tests. We hypothesize that it might be due to variation associated with the multiple epitopes targeted. We could further hypothesise that the reproducibility of pAb would have been even lower with multiple batches, considering reports in the literature that there are batch-to-batch differences from one animal to another for pAb [45].

It is generally considered that polyclonal antibodies will have higher apparent binding capacities than monoclonal or recombinant antibodies due to the variety of epitopes targeted compared with the single epitope target of a monoclonal or recombinant antibody [23,24,25]. The recAb that we employed was almost as efficient as pAb, so it is no doubt directed towards a dominant epitope that allows efficient capture. The increased reproducibility that we observed with recAb is no doubt a considerable advantage over pAb for use in biosensors, especially considering that its capture efficiency was almost as high as that of pAb. In the future, the use of multiclonal antibodies, a mixture of monoclonal recombinant antibodies with a completely defined composition, further promises to provide detection reagents that combine the advantages of both recombinant and polyclonal antibodies [45].

## 4. Conclusions

With the ubiquity of *L. pneumophila* in water sources, the development of an effective biosensor is imperative to improve prospective and preventive action toward outbreaks. We report two important steps towards attaining this objective. A new recombinant antibody targeting the LPS of *L. pneumophila* spp1 has been developed [28], which we have evaluated in the development of biosensors for the capture of whole *L. pneumophila*. We have also developed a stringent washing protocol that maintains the viability of *L. pneumophila*. Polyclonal and recombinant antibodies have been shown to specifically capture large targets, such as whole bacteria, and this opens the door to multiple biosensor approaches where any of the components of the bacteria can be measured for detection or characterisation, such as we have used with PCR detection of bacterial DNA sequences.

Through the advancement of biotechnology, reproducible production processes have been developed to manufacture recAb in order to forgo the use of animal and pAb production with their inherent variation and animal welfare concerns. Although they target only single epitopes, we have demonstrated that recAb are capable of capturing intact target bacteria. The capture of *L. pneumophila* at different concentrations by recAb and pAb in different environments showed comparable sensitivity and specificity. However, the lower standard deviations, indicating a more reproducible capture, favour recAb, even compared with a unique batch of pAb. For biosensors, precision is crucial in order to obtain constant, reproducible, and stable results for repeated tests.

As these results are in favour of the use of recombinant antibodies under pristine laboratory conditions, it is still necessary to confirm these results with cooling tower water or other environmental water samples using surfaces of different biosensors, such as those based on gold surfaces or on digital photocorrosion of GaAs/AlGaAs nanoheterostructures.

## Figures and Tables

**Figure 1 biosensors-12-00380-f001:**
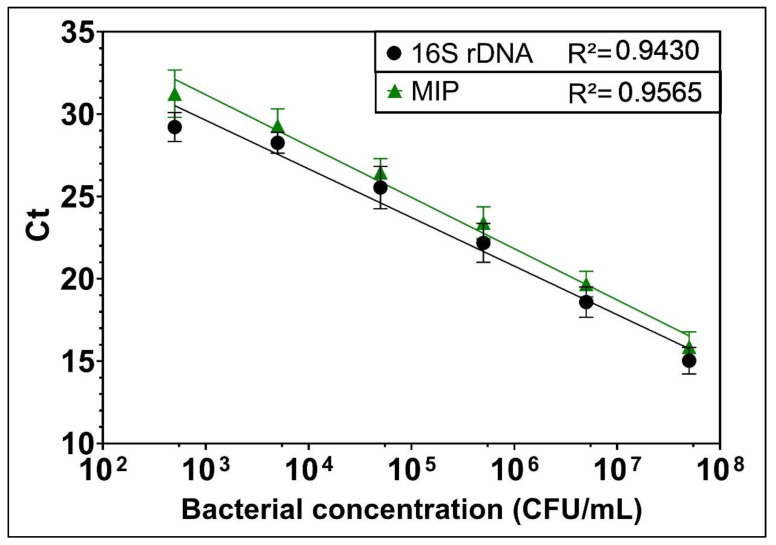
Correlation of Ct values derived from second derivative of PCR amplification using 16S ribosomal DNA universal bacterial primers and MIP gene primers specific for *L. pneumophila* with bacterial suspensions at different concentrations in microplate wells (*n* = 15). The results were fitted for MIP and 16S DNA as Ct = 40.532 ± 3.117 × log(10) and 38.487 ± 2.953 × log(10), respectively. Error bars are shown to indicate standard deviation.

**Figure 2 biosensors-12-00380-f002:**
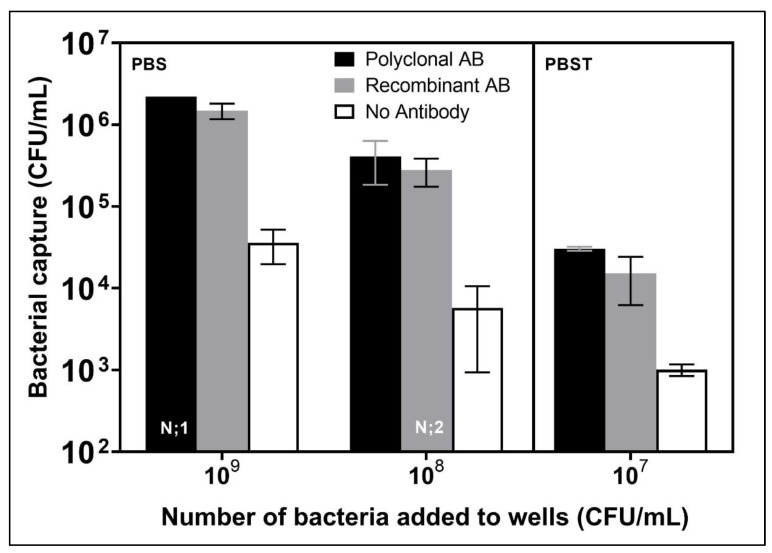
Number of bacteria captured per well with pAb (Black), recAb (grey) or without Ab (white) for different concentrations of *L. pneumophila* added to wells (*n* = 3, except where indicated differently on the bottom of the bar). The results were quantified by qPCR (MIP primers). Error bars are shown to indicate standard deviation.

**Figure 3 biosensors-12-00380-f003:**
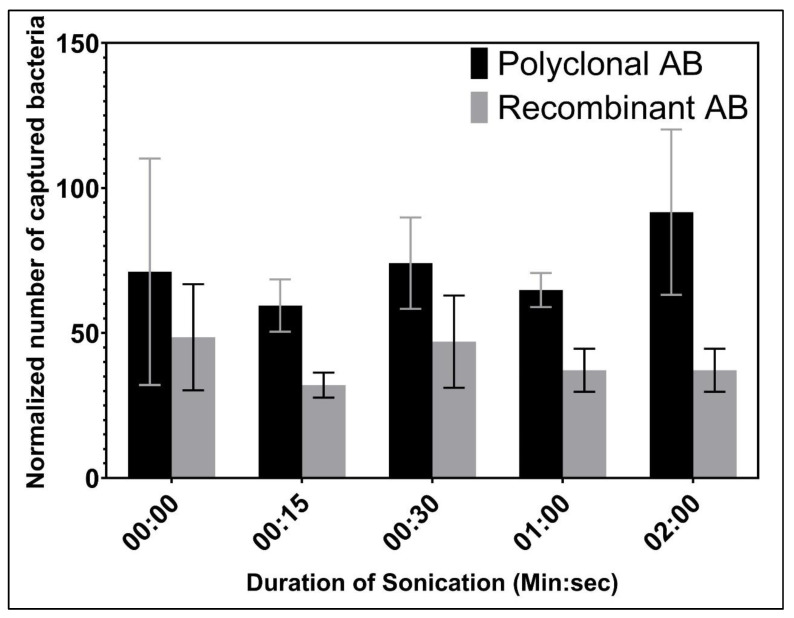
Normalized number of bacteria captured in wells coated with different pre-sonicated Ab and then exposed to *L. pneumophila* at 10^8^ CFU/mL (*n* = 3). Capture was quantified by qPCR (MIP primers) and normalized relative to control wells not coated with Ab (background value) (*n* = 2). Error bars are shown to indicate standard deviation.

**Figure 4 biosensors-12-00380-f004:**
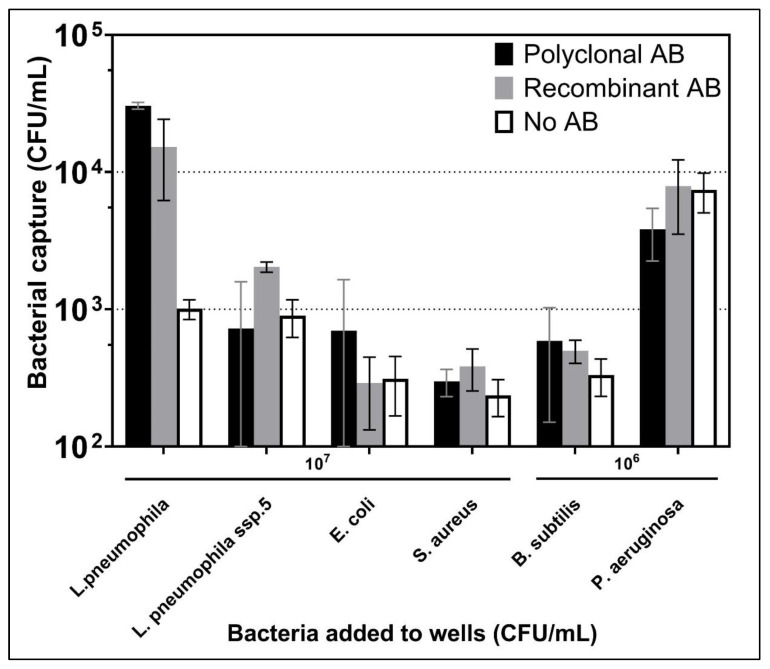
Capture of different bacteria in wells coated with pAb or recAb. The concentration of bacteria added to the wells was 10^7^ CFU/mL for *L. pneumophila*; *E. coli*; *S. aureus*; *L. pneumophila* ssp 5 and 10^6^ CFU/mL for *B. subtilis*; *P aeruginosa* (*n* = 3). The bacterial capture levels were quantified by qPCR using 16S rDNA primers and probe. Error bars are shown to indicate standard deviation.

**Figure 5 biosensors-12-00380-f005:**
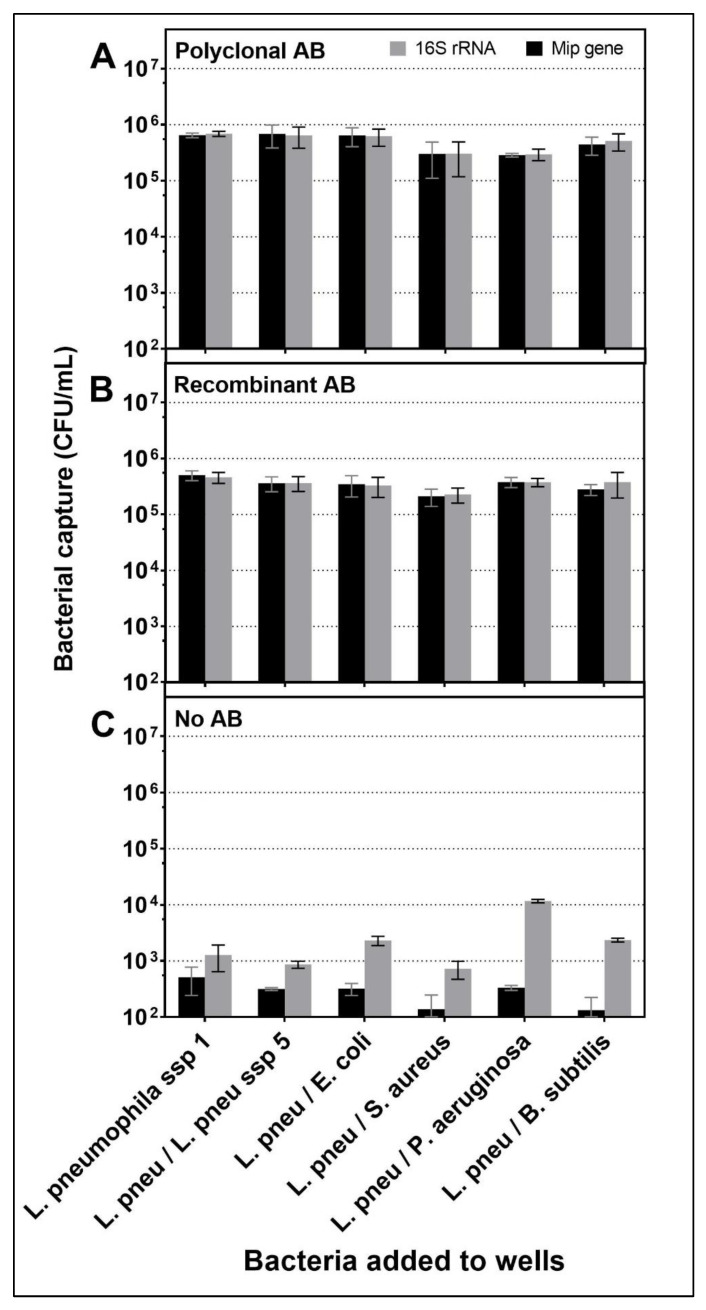
Bacteria captured in wells coated with (**A**) pAb, (**B**) recAb, and (**C**) control no Ab, exposed to various suspensions of live bacteria. *L. pneumophila* ssp 1 at 10^7^ CFU/mL was used alone or mixed with other bacteria at 10^7^ CFU/mL for *E. coli, S. aureus, L. pneumophila* ssp 5 (ratio 1:1) and 10^6^ CFU/mL for *B. subtilis* and *P. aeruginosa* (Ratio 10:1) mix (*n* = 3). Capture was quantified by qPCR with 16S rDNA universal primers (grey) or MIP primers (black). Error bars are shown to indicate standard deviation.

**Figure 6 biosensors-12-00380-f006:**
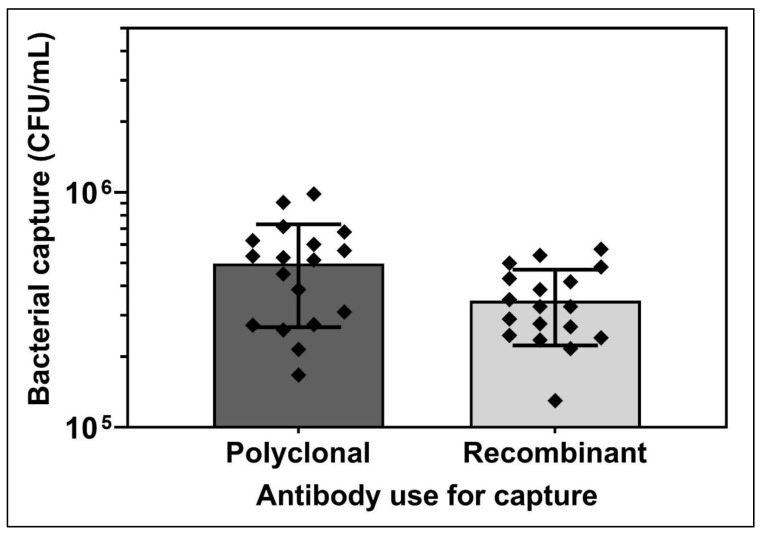
Compilation of number of *L. pneumophila* captured in wells coated with pAb (**left**) and recAb (**right**) exposed to 10^7^ CFU/mL *L. pneumophila* from Figure 5. Data were quantified with qPCR by employing 16S rDNA universal primers. Error bars are shown to indicate standard deviation.

**Table 1 biosensors-12-00380-t001:** Taqman PCR primers and probes.

Bacteria	Name		Sequence	Absorption/Emmision
*Legionella*	Mip			
*pneumophila*		Forward	5′-TTGTCTTATAGCATTGGTGCCG-3′	
		Reverse	5′-CCAATTGAGCGCCACTCATAG-3′	
		Probe	5′-(6-FAM)-CGGAAGCAA(ZEN)TGGCTAAAGGCATGCA–(IABkFQ)-3′	495/520 nm
Universal	1 rDNA			
		Forward	5′-TGGAGCATGTGGTTTAATTCGA-3′	
		Reverse	5′-TGCGGGACTTAACCCAAC A-3′	
		Probe	5′-(5CY5)-CACGAGCTGACGACARCCATGCA–(3BHQ_2)-3′	646/520 nm

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
