# Peer review of "Investigation of Conditions for Capture of Live Legionella pneumophila with Polyclonal and Recombinant Antibodies"

_biosensors, 2022, doi:10.3390/bios12060380_

Round 1

Reviewer 1 Report

In this manuscript, a comparative analysis was made of approaches that use polyclonal and recombinant antibodies for L. pneumophila detection by Paladines and co-workers. Similar studies have been discussed and compared more than 10 years ago (1-3). The authors should not just compare the differences between the two Abs on the basis of experimental conditions. They should also go through the binding constants between the two and the target in detail. In its current manuscript, I'm not sure what this article's unique contribution is. There are no unexpected outcomes.

References

1.Le Berre, Marie, and Marian Kane. "Biosensor‐Based Assay for Domoic Acid: Comparison of Performance Using Polyclonal, Monoclonal, and Recombinant Antibodies." Analytical letters 39.8 (2006): 1587-1598.

2.Alcocer, Marcos JC, et al. "Properties of polyclonal, monoclonal, and recombinant antibodies recognizing the organophosphorus pesticide chlorpyrifos-ethyl." Journal of agricultural and food chemistry 48.9 (2000): 4053-4059.

3.Lee, Mi-Gyung, et al. "Enzyme-linked immunosorbent assays of zearalenone using polyclonal, monoclonal and recombinant antibodies." Mycotoxin Protocols. Humana Press, 2001. 159-170.

Author Response

Thank you for your evaluation. I have added 2 of the references you provided to support our introduction where we say that recombinant and monoclonal antibodies have been used to capture small targets and are shown to be very efficient. We consider that our article shows that polyclonal and recombinant antibodies can also specifically capture large targets like whole bacteria and this is, in our view, a significant finding as it opens the door to multiple biosensor approaches where any of the components of the bacteria can be measured for detection or characterization such as we have used with PCR detection of bacterial DNA sequences. We also show that washing with 0.05% (and even 0.1%) Tween 20 does not kill the bacteria and thus the captured bacteria can be further studied for secondary characterization or to determine, for example, antibiotic sensitivity, We carefully described this in our cover letter, but see that this was not sufficiently emphasized in the manuscript abstract nor conclusions, so we have attempted to improve on this and thank the reviewer for pointing out that we had not clearly emphasized our article’s unique contribution! We would have liked to provide binding constants for these antibody target reactions, but our detection method is an endpoint analysis based on secondary analysis of the bacteria using PCR and does not allow measurement of binding constants. Indeed we expect that 2 “binding” constants would be significant: the first between the antibodies and the target antigen; and the second involving the binding strength of the antigen to the bacteria (the force necessary to extract the antigen from the bacterial surface).  

Reviewer 2 Report

The present study is devoted to comparing the use of immobilized polyclonal antibodies and recombinant antibodies as potential binding components for biosensor detection of Legionella pneumophila.
As a main result, it is noted that polyclonal antibodies have a higher binding capacity, but are characterized by lower reproducibility. However, in some experiments, polyclonal antibodies show a smaller standard deviation than recombinant antibodies (eg Figure 4 first bars). Please explain this.
It is also necessary to give more information regarding polyclonal antibodies: are these antibodies purified with G protein? If so, then most of the immunoglobulins of the polyclonal antibody preparation (more than 90%) are general immunoglobulins and do not bind to the antigen. This means that, in terms of active molecules, polyclonal antibodies are characterized by a significantly higher affinity than recombinant antibodies. What could be the reason for this?

Author Response

Thank you for your evaluation. Regarding the variation in standard deviations, we expect that biological experiments will give variable results and so we perform multiple tests and then measure standard deviations from these combined multiple tests to give a more appropriate evaluation.

According to the specifications from the supplier of the polyclonal antibodies, they are IgG that have been raised against whole cells of Legionella pneumophila and purified according to methods developed by the supplier. Do I understand correctly that the reviewer is raising the point that natural antibodies directed against LPS are predominantly IgM (because class-switching does not occur in antibodies to non-protein antigens) and would not be found in large amounts in a “general” (IgG) preparation? As such, the polyclonal antibodies are probably directed primarily towards both internal and surface proteins, but using them in a coating buffer at 100 µg/mL probably still provides sufficient antibodies. Indeed, this does not appear in our study to prevent them from efficiently and specifically capturing L. pneumophila.

Reviewer 3 Report

This paper investigated the possible application of polyclonal antibodies versus recombinant antibodies in capturing live Legionella pneumophila from water samples. Although the manuscript is carefully written and the results are logically presented, it is not clear which method is most efficient. The manuscript should be subjected to major revision:

Figure 1: the slopes should be negative. The authors must explain why the trendlines are so similar.

Figure 2: the non-specific adsorption at high exposure concentrations was relatively high reaching approximately 70% of the probe signal. What was the efficiency of pAB and rAB at lower concentrations? These results must be discussed.

More details about the capturing ability of pAB and rAB should be given at lower bacteria concentrations.

Overall, it is not clear if one method is more efficient than the other one. Although the authors attempted to make a comparison among them, the experimental results seem to be comparable.

Conclusions: the reference reported should be 24.

Author Response

Thank you for your evaluation.

In Figure 1 the slopes are supposed to be negative, because the Ct (cycle threshold) of PCR reactions decreases with increasing concentrations of target. PCR reactions are quite reproducible and similar targets will therefore give very similar slopes as we see here.

In Figure 2, we calculated that non-specific adsorption at high exposure was only 2 to 3% and at 107 was 6.6% compared to levels of specific capture. Already with exposure of 107 bacteria to wells without antibodies, we observed capture of 1X103 bacteria which is only 2 times the lower limit of detection of bacteria by PCR, so we were unable to provide meaningful results for capture of bacteria in wells without antibodies at lower concentrations. With pAB and recAB coated wells, we observed similar percentages of captured bacteria to those shown for higher exposure levels.  

We thank the reviewer for pointing out that we had not clearly emphasized our article’s unique contribution which we have emphasized in the Abstract and Conclusions. We prefer recAb, but agree that pAb is also an interesting solution for capturing whole, live bacteria, whereas many other authors have shown that due to its lower specificity, it is not ideal for immunofluorescent viral detection and other detection protocols for individual proteins or small targets.

Indeed in the Conclusions we reported the wrong reference which should be 26 and becomes 28 after the addition of 2 references suggested by reviewer 1.

Round 2

Reviewer 1 Report

I have no further comments.

Reviewer 2 Report

As far as I understood, the authors do not have sufficient information about the commercial preparation and methods for its purification. In this case, I withdraw my question about the ratio of specific and nonspecific antibodies in the preparation.

I believe that the manuscript may be published in the form presented.